# Genomics and Transcriptomics Reveal Genetic Contribution to Population Diversity and Specific Traits in Coconut

**DOI:** 10.3390/plants12091913

**Published:** 2023-05-08

**Authors:** Kobra Yousefi, Siti Nor Akmar Abdullah, Muhammad Asyraf Md Hatta, Kong Lih Ling

**Affiliations:** 1Department of Agriculture Technology, Faculty of Agriculture, Universiti Putra Malaysia, Serdang 43400, Selangor, Malaysia; yousefi.kobra@student.upm.edu.my (K.Y.); m.asyraf@upm.edu.my (M.A.M.H.); 2Laboratory of Sustainable Agronomy and Crop Protection, Institute of Plantation Studies, Universiti Putra Malaysia, Serdang 43400, Selangor, Malaysia; lihling@upm.edu.my

**Keywords:** *Cocos nucifera*, genomics, transcriptomics, molecular markers, fruit development

## Abstract

Coconut is an economically important palm species with a long history of human use. It has applications in various food, nutraceuticals, and cosmetic products, and there has been renewed interest in coconut in recent years due to its unique nutritional and medicinal properties. Unfortunately, the sustainable growth of the coconut industry has been hampered due to a shortage of good quality seedlings. Genetic improvement through the traditional breeding approach faced considerable obstacles due to its perennial nature, protracted juvenile period, and high heterozygosity. Molecular biotechnological tools, including molecular markers and next-generation sequencing (NGS), could expedite genetic improvement efforts in coconut. Researchers have employed various molecular markers to reveal genetic diversity among coconut populations and for the construction of a genetic map for exploitation in coconut breeding programs worldwide. Whole genome sequencing and transcriptomics on the different varieties have generated a massive amount of publicly accessible sequence data, substantially improving the ability to analyze and understand molecular mechanisms affecting crop performance. The production of high-yielding and disease-resilient coconuts and the deciphering of the complex coconut genome’s structure can profit tremendously from these technologies. This paper aims to provide a comprehensive review of the progress of coconut research, using genomics, transcriptomics, and molecular markers initiatives.

## 1. Introduction

Coconut (*Cocos nucifera* L.) is a perennial palm found in the tropical and subtropical regions of the world. It plays a crucial role in the economy of several countries in the South and Southeast Asian regions, such as Indonesia, the Philippines, and India. It is called the tree of life, as all parts of this plant have beneficial uses and can be a source of income for farmers. However, many coconut plantations were planted one or two decades ago. The presence of old and senile coconut trees in these plantations, environmental stresses, and the lack of high-yielding varieties are significant concerns affecting the sustainability and growth of the industry. Coconut farmers also face major difficulties due to the unstable market for dried coconut kernel products, resulting in decreased farmers’ income [1]. Nevertheless, development in recent years has held great promise for the coconut industry. Coconut oil shows an increasing application in cosmeceuticals and personal care products. Its applications in the food industry is strengthened due to its unique and desirable physical, nutritional, and medicinal properties [2]. The renewed interest in coconuts is supported by the increasing trend in the world-wide production of coconut fruits. Global coconut production has increased steadily over the past two decades, from around 51 million metric tons in 2000 to an estimated 61.5 million metric tons in 2022 [3].

The coconut is a versatile and nutritious food item which has been part of human diets for generations. Coconut milk is rich in beneficial fats with an excellent taste, making it popular in curries, stews, and smoothies. Coconut milk and cream are also popular dairy-free alternatives to regular milk, cream, and butter in cooking. Coconut oil is another favorite coconut-derived product often used in cooking and baking, as well as personal care and cosmetic goods. Moreover, coconut water is a natural electrolyte-rich beverage often drunk after exercise or sickness to rehydrate and replace minerals [4,5]. The high concentration of medium-chain fatty acids (MCAF) in coconut oil is highly beneficial for human health. Coconut oil and kernels are antibacterial, antiviral, anti-diabetic, antimicrobial, anti-inflammatory, and protective of the heart. Moreover, the fiber in coconut endosperm has hypocholesterolemic qualities and its hemicellulose helps lower blood cholesterol. Besides these features, coconut water has antioxidant and antithrombotic effects that help to prevent the onset of illnesses such as diabetes, heart disease, and high blood cholesterol [2,6,7,8].

In the genus Cocos, only the species Cocos nucifera has survived—a diploid (2n = 32) with a large genome size of about 2.4 Gb, the largest among the palm species. It is traditionally propagated by seeds (nuts), and its reproductive maturity depends on the genotype and culture environment [9]. The dwarf varieties, which are able to bear fruit after three years, are mainly self-pollinating, while the tall varieties are cross-pollinating, taking at least six years to start fruit production. Hybrid cultivars (Tall × Dwarf) coconut have been produced by various interspecific crossings, such as Maypan (Malayan Yellow Dwarf × Panama Tall), which is known to be high yielding and resistant to the lethal yellowing disease [10]; MAWA (Malayan Yellow Dwarf × West African Tall), which was produced in order to obtain a high nut and copra yield [11]; and Matag (Malayan Red Dwarf × Tagnanan Tall), which is high yielding and has early maturity with large fruit [12]. In order to increase the productivity of a coconut farm, it is necessary to develop improved cultivars with high productivity that are resistant to pathogens and pests as well as compatible with specific growing environments [1]. Different cultivars and varieties of coconut have been produced over a long period of natural evolution and artificial selection. Thus, preserving the coconut’s genetic resources is an important goal in the breeding programs that aim to produce improved and new varieties of coconut. The production of high-yielding dwarf varieties that ease harvesting with uniform fruit and high nutritional value for consumption in the world markets is desirable [13,14]. The increase in the weight of meat or copra is significant for meeting the market demand and has been given much attention. The number of nuts is also important for sellers who often sell coconuts in numbers. However, the towering stature, hard seeds, slow reproduction rate, and long generation time of the coconut plant complicate breeding and selection efforts [15]. Breeding techniques have increased the number of nuts produced by dwarf–tall hybrids that outperform tall coconut cultivars by more than 45% [16]. The number of nuts per coconut tree more than compensates for the hybrids’ poor copra content compared to tall coconuts [17]. 

In the 1990s, molecular markers were introduced in coconut breeding programs. These markers were helpful in assessing coconut genetic variation and creating genetic linkage maps for marker-assisted selection (MAS) [18]. In addition, the molecular markers helped to identify and select coconut genotypes with favorable traits. In order to encourage molecular-based research in the palm family, Fan et al. (2013) used de novo assembly and RNA-seq technologies to analyze the coconut transcriptome from the major coconut-producing countries. Next-generation DNA and RNA sequencing approaches have produced large volumes of data relatively efficiently, serving as valuable resources for the genetic improvement of coconut [19,20]. These advanced sequencing techniques would enhance speed and precision in achieving the targeted results of coconut breeding programs through marker-assisted selection. The latest progress in molecular marker applications, whole genome and transcriptome sequencing, genetic linkage profiling, marker-assisted breeding, and genotyping by sequencing (GBS) [16] that advance coconut research and are beneficial to the coconut industry are discussed in this review.

## 2. Fruit Development in Coconut

Similarly to most tropical fruits, coconut has a long shelf life due to the thick covering of the fruit, making it suitable for export and processing. In most cases, yield is expressed as the number of nuts and copra produced per nut. It is challenging to achieve coconut yield enhancement because the number of nuts per bunch and the size of the nuts are negatively associated [18]. Thus, enhancing naturally existing coconut varieties via selection alone cannot improve both attributes simultaneously. Nut size, the weight of the whole nut, the nut without husk, the shell, and water content are possible additional yield metrics [1]. 

Coconut fruit development is a highly unique process; however, until now, only a few genes or developmental pathways related to the development of coconut fruits have been studied [21,22,23]. Many enzymes and genes have been linked to particular metabolic or biosynthetic pathways for carbohydrates, fatty acids, or secondary metabolites that play a significant role in forming coconut endosperm [22]. Understanding the molecular biological processes involved in fruit development is a crucial first step in improving the quality of fruits [21]. Different genes regulate fruit size, shape, and weight, and the irregular expression of these genes causes developmental issues [24]. Plant hormones and transcription factors play a major role in regulating these genes’ expression in developing climacteric and non-climacteric fruits [24,25]. Transcriptomic investigations in the coconut demonstrated that each fruit growth and ripening stage includes distinct gene activities [26,27,28]. Some genes found are transcription factors known to govern major changes in the coconut fruit’s endosperm growth and ripening, such as the MADS-box [22,28] and CnGATA20 [27,29] transcription factors.

Since coconut is a seed-propagated crop, the order of the processes that leads to fruit development is essential for establishing the fruit characteristics of any coconut varieties. The flowering, pollination, and fertilization processes that culminate in a fruit set are the events that lead to the development of fruits [30]. Fruit development is divided into three stages. The first stage is the ovary formation and the start of cell division, which is a fruit set. Cell division is the most prominent aspect of the second phase. The fruit grows during the third phase, mostly via cell expansion. The ripening begins after the fruit cells have completely grown and the fruit has developed [31]. 

The early fruit set of coconut (*Cocos nucifera* L.) determines the eventual production. The production of fruit sets in the first months after the emergence of inflorescences is one of the essential variables driving performance in coconut [32]. Thus, understanding the parameters affecting fruit sets is crucial for estimating and accurately predicting output [33]. Early fruit production in perennial plants, such as coconut and oil palm [34,35] are hampered by low light, high temperatures, and lack of water [36]. The number of fruits produced in freshly opened inflorescence may be limited under non-stressful conditions due to a lack of female flowers or intense competition for assimilates from already formed fruits (strong sinks) [36].

The coconut fruit from the tall plant variety begins to produce flowers at a monthly interval after five to six years of planting. The fruit development process starts immediately following the fertilization process. The expansion of the embryo sac occurs throughout the fruit development process, while in the center, a vast vacuole is formed [37]. The endosperm of coconuts is the primary edible tissue for fresh eating and processing, making it the most significant component of the fruit. The cavity is filled with nut water or liquid endosperm as the nut volume expands in the first six months. Subsequently, the endosperm solidifies from the other end of the stalk, gradually extending towards the inner area. The hardening of the solid endosperm to form a solid kernel consisting of 84% oil from the initial jelly-like form occurs between the seventh and ninth months. This process continues until the 13th month, when the firm white flesh of endosperm is formed owing to intracellular oil deposition. Differences in the fatty acid composition are revealed at different stages of endosperm development. Beginning around the fourth to seventh month of development, when the solid endosperm begins to form, the rate of fatty acid production rises steadily until the twelfth month [38,39]. 

According to research on the growth, maturation, and accumulation of granular matter in coconut fruits, [40] identified four distinct development stages: initial, pre-pollination development, post-pollination development, and ultimately, maturity and senescence. According to their findings, endosperm, shell, and husk development began in the sixth, fifth, and first months, respectively, with all three components developing concurrently from 5–8 months after fertilization. The rapid development periods of the husk, endocarp, and endosperm are extended from 3–7 months, 5–9 months, and 6–10 months, respectively—the variety and development circumstances influence nut weight and individual nut component properties [41]. The color of the coconut husk indicates its ripeness. Depending on the variety, dwarf coconuts are classified as green, yellow, or red based on fruit color [42]. However, in most tall coconut varieties, the peel of the fruit changes color from green to brown as it grows. An immature coconut is filled mainly with coconut water and has a bright green shell. The fruits reach peak maturity in tall coconuts when the flesh has solidified, and the entire outer husk is brown in color [43]. Figure 1 illustrates the pre-pollination and post-pollination stages in coconut fruit development.

## 3. Genetic Diversity Studies Using Molecular Markers

Research on coconuts had made great strides since the mid-1990s, when molecular marker technology was first used. Genetic markers are more informative than the conventional phenotypic data gathered in the field or greenhouse. Breeders have employed molecular markers to access the targeted genomic sites in various crops [19]. Molecular markers enable the accurate detection of varieties at the nucleic acid level and the precise distinction of genetic diversity from environmental diversity in annual and perennial plants. The technology was initially employed in coconut for a variety of purposes, such as genetic diversity [51,52], developing DNA fingerprinting [53,54], and more recently, creating linkage maps [55,56] and assessing hybrids [57,58]. Earlier efforts in the diversity analysis of coconut germplasm focused on using restriction fragment length polymorphism (RFLP), random amplified polymorphic DNA (RAPD), and amplified fragment length polymorphism (AFLP) markers, with an increasing tendency in the last decade to employ simple sequence repeats (SSR) [18]. 

The genetic diversity of coconut accessions was studied by employing polymerase chain reaction (PCR)-based molecular marker systems, including RAPD [59,60,61,62], inter-simple sequence repeat (ISSR) [63,64], and SSR [65,66,67,68,69,70]. With a total of 45 markers (15 from each marker system), the RAPD, ISSR, and SSR marker systems produced 82 bands. The different types of markers found a high proportion of polymorphic alleles and certain distinctive bands. In addition, the cluster similarity coefficient values for the different marker systems also ranged from 22% for RAPD to 83% for ISSR and from 50% to 97% for SSR. Based on these results, it was determined that all three molecular marker methods provided similarly accurate assessments of the genetic diversity of coconut genotypes [71]. 

Due to the highly heterozygous character of the coconut, due to its out-breeding pollination behavior, SSR markers have proven particularly useful for genotyping. SSR markers successfully gave unambiguous information on the genetic relationships and identification of representative collections, hybridity tests, finding somaclonal variants in tissue-cultured coconuts, and constructing linkage maps [18]. Genetic diversity among 48 coconut trees in Kenya’s lowland coastal zone was analyzed using SSR marker technology [67]. Genetic diversities were measured using data from 15 SSR marker loci using Popgene version 1.31, and they ranged from 0.0408 at marker locus CAC68 to 0.4861 at marker locus CAC23, with a mean of 0.2839. Aside from discovering polymorphic marker loci for Kenyan germplasm, the study also found that the within-population variation was substantial at 28%, while variation across populations was minimal at 2%, indicating that molecular variation is independent of where plants are grown. Geethanjali et al., 2018, investigated the population structure and genetic diversity of 79 coconut accessions using 48 SSR loci. The genotypes tested demonstrated a reasonably high level of genetic diversity and the presence of an organized population depending on their geographical origins [70].

Four coconut accessions—giant, mini, micro, and ordinary—were examined for population structure and genetic diversity. Nineteen SSR markers were used to study these accessions in Minicoy Island, India [72]. A total of 70 alleles were found, with a mean of 3.68 alleles found per locus. According to the findings, Laccadive Mini Micro Tall is a genetically unique accession. Five native accessions from the Konkan region of Maharashtra, India—the “Banawali”, “Gangabondam Green Dwarf”, “Pratap”, “Konkan Bhatye Coconut Hybrid-I”, and “East Coast Tall”—were characterized using 14 SSR markers by Rasam et al. (2016) [73]. The levels of polymorphism observed using SSR markers varied from 85.7% to 100% across the five accessions. Genetic diversity was found in the five accessions, both across and within accessions. Thus, the application of SSR has been very successfully employed for diversity studies within and between populations and accessions. Its utilization has been successfully extended in marker-trait association studies [70,71,72,73,74].

Point mutations result from single nucleotide polymorphisms (SNPs), which account for the vast genetic variances between individuals. A large quantity of sequence data from the parent genotypes are required to develop usable SNP markers, since all species have vast stores of dormant SNP markers [74]. Due to their steady distribution throughout the genome, low mutation rates, and the accessibility of simple detection techniques, SNPs offer certain benefits over alternative marker systems. Exonic SNPs, also known as cDNA SNPs, are SNPs that are located inside exons. However, many SNPs have been discovered to exist in intronic regions of the genome, where they are part of a subgroup of non-coding SNPs known as ncSNPs. SNPs in promoter regions (pSNPs) are indicative of a strong effect of SNPs on the activity of related genes [75]. 

Research on Brazilian dwarf coconut populations using RAD-sequence tags found that SNP markers are highly sensitive to genetic diversity [76]. Five loci, including genes involved in reproduction, respiration, and defense, were identified by Muoz-Peréz et al. (2020) through a population structure analysis and the genetic diversity of planted cultivars using SNPs markers [77]. Pesik et al. (2017) set out to analyze the WRKY gene nucleotide sequence diversity in an Indonesian Kopyor (endosperm mutations leading to intra-shell endosperm breakage) coconut germplasm. Primers targeting eight relevant SNPs were constructed, and duplex PCR was useful for distinguishing among five Kopyor coconut varieties [78]. Muñoz-Pérez et al. (2022) identified SNPs from the coconut genome based on genotyping by sequencing (GBS) to analyze the genetic diversity, population structure, and the linkage disequilibrium (LD) of a diverse coconut panel consisting of 112 coconut accessions from the Atlantic and Pacific coasts of Colombia. A total of 19,414 of 40,614 SNPs were anchored to chromosomes. Among these, 10,338 were found only in the Atlantic population, 4606 in the Pacific population, and 3432 could distinguish between the two populations. A clear population structure at K = 4, distinguishing between Pacific and Atlantic coast accessions, as shown by a filtered subset of unlinked and anchored SNPs (1271). Nucleotide diversity was low, LD decay was sluggish, and the fixation index (Fst) was low in Pacific populations, whereas Fst was somewhat higher and LD decay was quicker in Atlantic populations [56].

### Linkage Mapping, QTL Identification, and Association Studies 

In coconut, the quantitative characteristics of interest include fruit quantity, weight, size, and features giving resistance/tolerance to stressors. Influenced by numerous genes, these features are referred to as quantitative, polygenic, multifactorial, or complex. Quantitative trait loci (QTL) are genomic areas that include the genes and genetic factors that drive these quantitative characteristics [79]. When pinpointing a specific chromosomal location, bi-parental mapping is the method of choice. However, due to the coconut’s perennial nature, creating a bi-parental population is time-consuming and labor-intensive. Genome-wide association studies (GWAS) and linkage disequilibrium analyses (LDA) exploit historical recombination phenomena within pools of unrelated individuals in order to identify the sequence variants producing phenotypic variation to identify QTLs in a collection of crop genotypes. Trait-marker association analysis and association mapping need a germplasm collection with high trait variability and no linkage disequilibrium between unrelated genes [70].

The creation of genetic linkage maps, which show the order and genetic distances of genetic markers along each chromosome, is a crucial step in genetic research. Marker co-segregation with QTLs discovered for agronomic characteristics may provide opportunities for marker-assisted selection in coconut breeding programs [80]. Association mapping, which involves linking quantitative features and markers in natural populations, is an alternate strategy for establishing segregating populations for linkage mapping. Association studies require a high concentration of molecular markers, necessitating a high-throughput marker system, such as diversity array technology [81]. Association mapping is based on linkage disequilibrium and the search for connections between closely connected markers. This mapping method would be very beneficial in a crop such as coconut to avoid the difficulties of creating traditional segregating populations for genomic mapping. However, marker saturation across the coconut genome is needed for the association mapping of QTL to enable marker-assisted selection in coconut [79]. 

Baudouin et al. (2006) conducted QTL analyses for fruit weights and ratios in a segregating offspring of a Rennell Island Tall (RIT) genotype to examine the genetic mechanisms affecting fruit components in coconuts. A better linkage map was created using one AFLP primer combination, eight SSR markers from related oil palm species, and forty-eight unique coconut SSRs. An RIT parent linkage map of 1849.8 cm was produced, with 16 linkage groups ranging from 51.9 to 181.8 cm. The linkage group has around 274 molecular markers, and the revised map included 47 new markers, including 44 co-dominant and highly polymorphic SSRs. A total of 48 QTLs were involved in the expression of fruit component attributes, with 34 of them grouped into six clusters, showing the pleiotropic impact [82].

Using AFLP, SSR markers and COS clones, Riedel et al. (2009) constructed a linkage map for cuticular wax in a population of 94 offspring from a cross between East African Tall and Rennell Island Tall plants. After accommodating 704 markers, the combined map was 2739 cm in length. Across the whole coconut linkage map, 46 QTLs were identified, all of which were associated with cuticular wax composition [83].

Zhou et al. (2020) utilized SSRs to perform an association study in order to discover high-quality fatty acid alleles in 80 coconut accessions representing six different populations. Endosperm derived from coconut accessions were separated into sub-groups with a high and low fatty acid contents, and eleven SSR loci related to fatty acid content were found. The discovered alleles have significant phenotypic effects, with CnFatB3-359 and CnFatB2-830 exhibiting favorable and unfavorable effects, respectively [84]. Table 1 provides some information of association mapping analysis for fatty acid content in coconut. 

Because of the crop’s perennialism and extended juvenile stage, the size of the coconut genome for assembly, and the limited high-quality reference genomes, genome-wide association studies in coconut are very restricted. Yet, owing to the considerable difficulties in generating population mapping, the fact that the findings of association analysis and QTL mapping are correlated suggests that association mapping might be a helpful alternative technique for trait mapping in coconut [70].

## 4. Whole Genome Sequencing Efforts

Examining the coconut genome has enabled researchers to identify genes responsible for key traits, such as disease resistance, yield, and fruit quality. This knowledge is highly beneficial for developing novel coconut varieties that are more adaptable to changing environmental conditions and are more resistant to diseases that endanger coconut production. Researchers now have an effective tool for studying coconut biology and creating new technologies that may help assure the sustainability of coconut production owing to the accessibility of the sequences of the coconut genome. The coconut genome is vast and has intraspecific variation linked to domestication, with tall cultivars exhibiting much more variance than dwarf cultivars [85]. According to Alsaihati et al. (2014), the coconut genome comprises between 50 and 70% of repetitive sequences [86]. Evolutionary research suggests that fluctuations in sea level throughout the Pleistocene glaciations were responsible for the massive expansion of transposable elements in the coconut genome [87]. Based on the annotations from other species, several similar genes and their possible participation in coconut biological processes have been found. Embryogenesis, general plant growth, fruit development, and fruit nutrition quality are just a few of the many significant biological processes discovered [88]. 

The whole draft genome of the coconut (Hainan Tall variety) was made accessible by de novo sequence assembly. It included mapping data for six QTLs and 97 kb of association mapping data, for a total of 2.42 Gb [9]. It is 34% bigger than the oil palm genome (1.8 Gb) [89] and 3.6 times larger than the date palm genome (0.67 Gb) [90]. A total of 419.67 Gb raw reads were produced by Illumina HiSeq 2000 using a mix of paired-end (PE) and mate-pair (MP) libraries to achieve 173.32 read depth coverage of the expected genome of 2.42 Gb and an estimated depth of 173. The N50 value from this preliminary genome sequencing project was 418 Kb, while the scaffold length was 2.2 Gb [9]. According to the annotated coconut genome results, long terminal repeats (LTRs) make up the vast majority of the TEs present in the genome. Compared to the similar palm species *Phoenix dactylifera* [91] and *Elaeis guineensis* [89], the number of predicted genes in the coconut genome (28,039) is much lower. Based on an examination of their evolutionary relationships, the 119 anti-porter genes discovered in the coconut genome may be classified into 12 distinct functional classes, similar to those identified in the genome of the model plant Arabidopsis. [9].

Using Illumina Miseq, PacBio SMRT, and Dovetail Chicago technologies, the whole genome of the cultivar ‘Catigan Green Dwarf’ (CATD) was sequenced [92]. The overall length of the completed genome assembly (N50 = 570,487 Kb) comprising 7998 scaffolds was 2.1 Gb (97.6% of the projected genome size of 2.15 Gb). This study identified 58503 SNP (compared to Hainan Tall genome) and isolated 7139 genetic SSRs from genes involved in oil production, biotic stress, and drought tolerance, while 25, 157, and 39 SSR markers, for example, are physically separated from characteristics linked with drought response, insect or pest resistance, and oil biosynthesis, respectively. Interestingly, based on a study of exon synteny and variations in *Arecaceae* genome sequences, it was deduced that palms may have had three rounds of whole genome duplications (WGD) throughout evolution [92]. These genetic markers have enormous potential for application in molecular breeding procedures for coconut varietal development.

According to Nair et al. (2004), a disease known as root wilt has been discovered to have a substantial effect on palms in India, especially those of the Chowghat Green Dwarf (CGD) type [93]. Thomas et al. (2015) reported that these palms are self-pollinated and display genetic consistency based on microsatellite genotyping [94]. This implies that the CGD genome is likely to have greater levels of homozygosity, making it ideal for sequencing and acting as a reference genome for the coconut. Muliyar et al., in 2020, sequenced the organelle and nuclear genomes of the CGD cultivar, which is more resistant to root (wilt) disease, using the Pacific Biosciences RSII and Illumina HiSeq 4000 platforms. This resulted in the identification of a 1.93 Gb draft genome distributed among 26,885 scaffolds. Sequencing and screening resulted in the assignment of 11,181 proteins to 13,707 genes. The genome was filtered through to uncover 112 nucleotide-binding-site and leucine-rich repeat (NBS-LRR) loci, representing six types of resistance genes. The CGD genome contains sequence variance associated with disease resistance, making it a helpful tool for developing genomics-assisted breeding in coconuts [95]. 

Wang et al., 2021, used nanopore sequencing, high-throughput chromosome conformation capture (Hi-C), and Illumina technology to perform the high-quality sequencing of two representative coconut individuals from *Cocos nucifera* Tall (*Cn. Tall*) and *Cocos nucifera* Dwarf (*Cn. Dwarf*). The genomes of *Cn. tall* and *Cn. dwarf* were 2.4 and 2.39 Gb, respectively. In *Cn. tall* and *Cn. dwarf*, 29,897 and 28,111 protein-coding genes were predicted and annotated, respectively. The assembled contig N50s for the *Cn. tall* and *Cn. dwarf* ecotypes were 2.93 and 14.29, respectively, which is higher than previously reported coconut genomes [9,93]. A comparison of the genomes of *Cn. tall* and *Cn. dwarf* indicated that the two coconut ecotypes were separated by 2 to 8 Mya and preserved. At 47-53 Mya, an *Arecaceae*-specific whole-genome duplication (ω WGD) event occurred. This research used multi-omics analysis to determine the genetic basis of phenotypic differences between two coconuts. The copy number of GA-20 oxidase (GA20ox) and a single-nucleotide change in the promoter together play an important role in differentiating tall and dwarf height in coconut. Consequently, the variation in the height of the coconut plant is caused by a shift in gibberellin metabolism [96].

Using a backcross mapping population [MYD * (MYD * WAT)], Yang et al. (2021) demonstrated that scaffolds generated from the genome sequence could be placed on the sixteen pseudochromosomes, leading to the systematic placement of 77% of the genes onto sixteen linkage groups. Segregation distortion was seen on chromosome Cn15, as shown by a drastic decrease in heterozygotes (average heterozygosity of 27.5% vs. the anticipated 50%) due to solid selection among pollen grains favoring the maternal allele. In addition, RNA-seq data offer further information on the coconut osmotic adjustment signaling pathways, which remove reactive oxygen species while restoring Na^+^/K^+^ balance and stomatal closure in response to salt stress [87]. Table 2 provides the statistical data comparing the four nuclear genome assemblies of coconuts discussed above.

### 4.1. Discovery of Genes Involved in Oil Biosynthesis from the Whole Genome Sequence

Increasing the concentration of MCFAs and short-chain fatty acids is one of the main goals of coconut breeding in order to create remarkable coconut types with increased copra oil quality. One hundred and forty-four distinct palm species produce oils in their fruits, and numerous palm species lack oil [97]. As a result of the discovery of the whole genome sequences of date palm [91], coconut [9], and African oil palm [89], there was a strong desi to examine and compare the genetic components of oil biosynthesis pathways in these three related species [98]. Differences in oil production characteristics across these species provide the basic motivations for comparison. The mesocarp of an African oil palm produces oil, in contrast to the mesocarp of a date palm, which contains no oil. Oil from the kernel of the African oil palm, similar to that found in coconuts, is rich in lauric acid, which is an MCFA. On the other hand, date palm seed oil is mostly oleic acid and has a much lower oil content [98]. A total of 806, 840, and 848 lipid-related genes were found in *C. nucifera*, *E. guineensis*, and *P. dactylifera*, respectively. These genes were assigned to one of twenty-two lipid metabolism pathways [98]. The bulk of these lipid-related genes were substantially similar across the three species studied. Coconut and oil palm, two oil-producing species, shared many of the most critical genes in the lipid and carbohydrate metabolic pathways but date palms varied.

For the biosynthetic process of general seed oil, Manohar et al. (2019) identified six potential genes involved in oil and MCFA production. Coconut genome assembly was built using 15× PacBio^®^ and 50× Illumina MiSeq sequencing reads of the CATD coconut variety [92]. Following an alignment of sequences, they were able to pinpoint the scaffolds that housed the putative genes encoding the fatty acid biosynthetic genes. Both evidence-based and ab initio prediction techniques were used to determine the 3D structures of these genes. Both coconut β-ketoacyl-ACP synthetase (KasII and KasIII) coding DNA sequences were described as the first report of these MCFA genes in coconut. The coding regions of each gene were targeted using custom PCR primers. EcoTILLING enhanced PCR settings for mining natural allele variations in 48 established coconut types from the Philippines. In both the ‘West African Tall’ (WAT) and ‘Aguinaldo Tall’ (AGDT) cultivars, a single nucleotide polymorphism (SNP) was found in the lysophosphatidic acid acyltransferase (LPAAT) genes [99]. The cloning and sequencing of a partial LPAAT gene in WAT and AGDT allowed for its characterization. In genomics-assisted coconut breeding for superior oil-producing variants, the discovered SNPs might serve as the basis for creating highly sensitive DNA markers for the high-throughput screening and selection of beneficial alleles.

*Elaeis guineensis* and *Phoenix dactylifera* homologous genome portions include around 73% and 62% of the genes involved in lipid synthesis found in coconuts, respectively [98]. Although the oil palm and coconut have diverged through time, resulting in significant differences in the gene expression pattern of fatty acid biosynthesis and TAG conversion, both species utilize the same conserved oil biosynthesis pathways to create and store oils in their respective tissues. Comparing these three-palm species’ lipid-related genes to Arabidopsis revealed that 25% of the genes are single copies [98]. In contrast, 50–55% of the genes have two or three orthologous gene copies for every gene in *Arabidopsis thaliana*, and despite the three palms sharing the structural properties of cis-acting regions, lipid-related genes exhibited minimal or no similarity in expression patterns. It was thus deduced that the sub-functionalization of various gene families, such as transcription factor (WRI1) and acyl-ACP thioesterase (FatB), occurred long before speciation.

### 4.2. Sequencing of Mitochondrial and Chloroplast Genomes

Understanding the coconut’s genetic diversity and evolution is critical for breeding efforts that aim to improve crop productivity and resilience to biotic and abiotic challenges. The genomic sequencing of organelles has offered information on the history of coconut migrations throughout various areas and continents. Moreover, the research on mitochondrial and chloroplast DNA has aided in identifying and authenticating coconut varieties, which is critical for conserving and maintaining this valuable crop.

Huang et al. (2013) sequenced the coconut chloroplast genome for the first time in its entirety. Similar to other angiosperm genomes, the coconut cp genome has a quadripartite, spherical molecule with a conserved structure and content organization. The total length of the chloroplast genome was 154.731 bp, with a G–C content of 37.44%. It contained 84 protein-coding genes, 8 ribosomal RNA (rRNA) sequences, 38 transfer RNA (tRNA) sequences, and 2 pseudogenes. Although the coconut cp genome’s basic gene organization, gene content, and repetitive structure are comparable to those found in other palm species, there are notable differences. A notable increase in RNA editing sites and the pseudogenization of rps19-like genes are two distinguishing features [100]. According to a study of two main commercially significant *Arecaceae* woody palms, oil palm [101] and date palm [102], the cp genome of the coconut is the shortest. All these palms have AT-rich cp genomes. The position of *C. nucifera* as the most closely related species of *E. guineensis* within monocot subtrees was well supported by phylogenetic analysis based on 47 chloroplast protein-coding genes from diverse plant groups.

Aljohi et al. (2016) sequenced the mitochondrial genome comprising 678,653 bp. It was determined as encoding 72 proteins, 9 pseudogenes, 23 tRNAs, and 3 ribosomal RNAs, with a G-C composition of 45.5% [103]. Contrary to the date palm mt genome, where it was discovered that 93.5% of the genome sequence was obtained from the chloroplast cp, the cp-derived sections of coconut accounted for 5.07% of the overall assembly length [104]. These included 11 tRNAs, 2 pseudogenes, and 13 proteins. The proportion of repetitions in the coconut mt genome is relatively high (17.26%) and includes both inverted (palindromic) and forward (tandem) repeats. Sequence variation analysis reveals that, compared to the nuclear genome, the mt genome’s transition/transversion ratio of 0.3 was substantially lower [104]. 

## 5. Transcriptomics Initiatives

Traditional coconut breeding projects across the globe have, for decades, focused on selecting traits such as copra production, delicate nut water quality, disease resistance, and drought tolerance. Nonetheless, employing NGS technology may allow for a better understanding of the molecular mechanisms associated with the different biological processes to enable knowledge-guided crop improvement that increases precision for a more desirable outcome. RNA-seq is a powerful NGS platform that enables the global analysis of genes expressed in a particular tissue and at specific developmental stages. It provides valuable data on differentially expressed genes, enabling comparison between the tissues of different developmental and physiological statuses and between disease and normal tissues. These studies enable researchers to identify the ideal settings or stages of development for maximal gene expression, offering essential guidance for future crop improvement. RNA-seq requires bioinformatics analyses from the initial sequence data clean-up, followed by alignment to genome sequence, to decipher the roles of the expressed genes in influencing biochemical pathways and the identification of differentially expressed genes.

### 5.1. Comparative Transcriptomics between Different Genotypes

Comparative transcriptomics allows scientists to compare gene expression in different coconut varieties or environments. It can aid in identifying genes responsible for traits such as disease resistance or fruit quality and develop new coconut varieties with improved characteristics or strategies to mitigate environmental stressors on coconut production. Using sequencing data produced from an Illumina transcriptome profile, 30 unique microsatellite markers were developed and used to examine 30 samples of coconut, 12 from China and 18 from Southeast Asia [105]. Allelic polymorphism was found to exist between accessions to varying degrees. The Chinese accessions were shown to represent a genetic subset of the Southeast Asian coconut accessions. Since the examined Southeast Asian accessions were similar to the Chinese ones, the scientists concluded that the two sets of accessions had evolved together. Population structure studies and historical records support the idea that ocean currents spread coconuts to China’s Hainan Province [105].

Aromatic Green Dwarf (AROD) is a one-of-a-kind accession endemic to Thailand that is mentioned in Figure 2 [106]. The liquid endosperm of this accession has a characteristic ‘pandan-like’ fragrance, and Saensuk et al. (2016) used transcriptome sequencing to discover the genetic basis for this feature. The aroma of coconut and fragrant rice is caused by the biological molecule 2-acetyl-1-pyrroline (2AP). The transcriptome assembly and transcriptome analysis of the Aromatic Green Dwarf liquid endosperm transcriptome revealed the rice orthologous gene(s) were involved in synthesizing 2AP. It was found that the length of transcripts expressing 2AP differed between aromatic and non-aromatic Green Dwarf genotypes, which could potentially explain the difference in the fragrant characteristics between them [107].

Kalpatharu is a tall, high-yielding variety that was produced by selecting high-yielding palms from the Tiptur tall coconut population, a renowned Karnataka cultivar in India (Figure 2). Kalpasree is a superior, root (wilt) disease-resistant dwarf coconut type with high yield potential for cultivation in homesteads in root (wilt) disease-prone areas, developed by selection from the indigenous dwarf cultivar, Chowghat Green Dwarf [108]. Coconut seedlings of the cultivars Kalpasree and Kalpatharu were exposed to soil water-deficit regimes to explore the molecular response of coconut to water-deficit stress. Biochemical, physiological, and growth analysis indicated variations in enzymatic antioxidants, lipid peroxidation status, and water consumption efficiency across the coconut genotypes [109]. The total plant water consumption efficiency in Kalpatharu (4.06) was much lower than in Kalpasree (4.74). Under high stress (25% ASM), Kalpatharu (5.68) outperformed the dwarf variety Kalpasree (3.84) in terms of water use efficiency (WUE). Using paired-end RNA-Seq, the leaf transcriptome profiles of control and water-stressed seedlings were studied. In all, 7300 genes were discovered to be differentially expressed in seedlings under water-deficit stress vs. control. However, there was an increase in transcripts encoding polyamine oxidase, arabinose 5-phosphate isomerase, and other enzymes, as well as a decrease in the aquaporin PIP1-2 gene in Kalpatharu leaves. Figure 2 shows a representative of the different coconut varieties with different phenotypic features. Transcriptomics have been used to explain the underlying molecular mechanisms based on differential expressed genes affecting different biochemical and physiological processes associated with the different phenotypes.

**Figure 2 plants-12-01913-f002:**
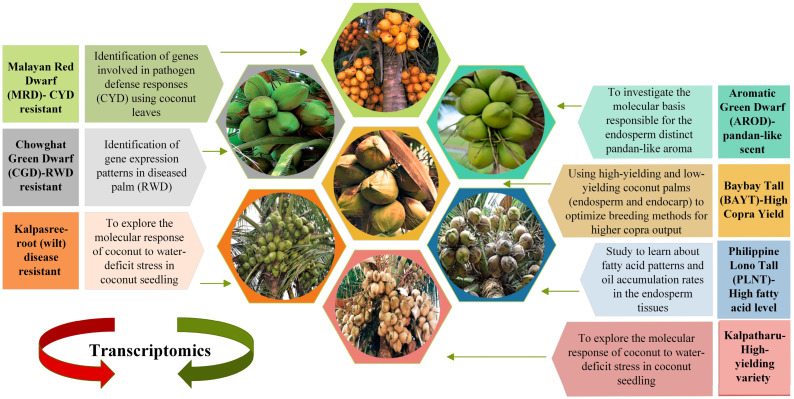
Distinct coconut varieties. The figure displays images and names of seven coconut varieties (MRD [110], CGD [111], Kalpasree [108], Kalpatharu [108], PLNT [112], BAYT [113], and Aromatic Green Dwarf) [114] investigated in various studies. Each variety is labeled with its corresponding name and briefly describes the research goal and investigation. Each variety shows the desired coconut fruit picture with an arrow. The scientific literature has explored various aspects of coconut varieties, including improving crop productivity, sustainability, and disease resistance. This figure serves as a visual reference to highlight the diversity and relevance of coconut varieties research, which can contribute to developing sustainable and innovative solutions for food and health.

### 5.2. Transcriptomic to Reveal Molecular Mechanisms Underlying Biochemical and Developmental Processes in Specific Tissues

Endosperm is a significant economic tissue in coconut. It is high in dietary fiber, vitamins, and minerals, such as potassium, magnesium, and calcium, making it beneficial for developing functional meals and supplements [115,116,117]. Moreover, the unusual composition of coconut endosperm (high quantities of medium-chain fatty acids, notably lauric acid) and its potential health advantages make it an appealing study subject in various sectors, including nutrition, medicine, and biotechnology [118,119,120]. A total of 57,304 unigenes were found in the transcriptome study of first emergent leaves and solid endosperm of the coconut variety Hainan Tall [121]. A large number of unigenes (23,168) were mapped to the Kyoto Encyclopedia of Genes and Genomes (KEGG) pathways, yielding 215 pathways involved in processes, which were as varied as galactose metabolism, plant-pathogen interaction, and signal transduction mediated by phytohormones. In addition, 347 unigenes involved in lipid metabolism and fatty acid production were identified. The processes represented by the unigenes in lipid biosynthesis include fatty acid biosynthesis, unsaturated fatty acid biosynthesis, the elongation of fatty acids, other lipid metabolism, and the citrate cycle. Twenty of these unigenes are especially noteworthy since they are involved in the biosynthesis of fatty acyl-ACP thioesterase. This important enzyme controls the fatty acid chain length by preventing carbon chain elongation. It was suggested that the relatively significant accumulation of medium-chain fatty acids (such as lauric acid) during endosperm development in coconuts was due to fatty acyl-ACP thioesterase overexpression [121]. The PLNT (Philippine Lono Tall) is a subvariety of the LAGT (‘Philippine Laguna Tall) with higher fatty acid (FA) and male endosperm (Figure 2). Ann Osio et al. (2019) published the subspecies’ first complete FA profile. Transcripts of RNA-seq were studied to learn about fatty acid patterns and oil accumulation rates in the endosperm tissues. Contigs (416,488 in total) were constructed from scratch, with 15,497 differentially expressed genes (14,356 upregulated and 1141 downregulated). It was found that endosperm growth, oil accumulation, oil biosynthesis, and cell membrane and cell wall biogenesis are affected by the differential expression of these unigenes. Understanding the molecular characteristics underlying the fatty acid production in coconut has been substantially aided by the different transcriptome analyses from coconut endosperm tissues [122].

Baybay Tall (BAYT) stands out among other tall coconut varieties in the Philippines for its exceptional copra yield (Figure 2). Punzalan et al. (2019b) undertook a differentially expressed gene study on the endosperm and shell of high-yielding and low-yielding coconut palms to optimize breeding methods for higher copra output. RNA was extracted from the endocarp and endosperm of nut tissues, and transcriptome sequencing was performed using Illumina HiSeq2000. Based on the functional analysis results, it appears that many transcription activators and regulatory proteins work together to speed up cell division, construct secondary cell walls, enhance energy metabolism, and activate the plant’s stress response, all of which contribute to the plant’s high nut production. Collectively, these activities boost kernel volume, thus increasing copra yield. From the nut tissues, 1945 genes were discovered to be differentially expressed. Potential gene-targeted markers (GTMs) based on 64 candidate genes were created and confirmed for use in the marker-assisted selection of high-yielding palms [123]. A transcriptome study was performed on developing gelatinous endosperm, mature embryos, and young leaves of a fragrant green dwarf coconut by Huang et al. (2014). A total of 58,211 unigenes, 61,152 unigenes, and 33,446 unigenes were found in the embryo, endosperm, and leaf tissues, after assembling the sequencing data. They found proteins in coconut that seem to be homologs of components needed for RNA-directed DNA methylation. These findings show that short RNA-mediated epigenetic control plays a significant role in coconut seed development, particularly in the mature endosperm [124].

Bandupriya and Dunwell (2015) classified key embryo-specific genes using transcriptome data from four EST libraries from early and late embryos, microspore-derived embryos, and adult leaves. The four libraries yielded 32,621 putative unigenes, 18,651 of which matched the non-redundant protein database. Comparing embryo-specific transcriptomes to leaf transcriptomes revealed significant differentially expressed genes such as chitinase, β-1,3-glucanase, ATP synthase CF0 component, thaumatin-like protein, and metallothionein-like protein [125]. Punzalan et al. (2019a) isolated total RNA from Laguna Tall (LAGT) nuts, leaves, and flowers at the mature stage. The data were then sequenced on an Illumina HiSeq 2000. A reference-free assessment approach for transcript abundance data was applied to test the correctness of each assembly. The combined transcriptomes generated 79,263 transcripts. RSEM produced 68,147 transcripts. Protein modification was involved in up to 33.8% of LAGT genes, according to gene ontology (GO) and KEGG categorization. The top 20 expressed genes were annotated using the nr database, indicating that the most highly expressed transcript is unique and new [126].

Using transcriptomics technology, Rajesh et al. (2016) first understood gene expression patterns during somatic embryogenesis in coconut. The transcript analysis of coconut embryogenic calli generated from the West Coast Tall Coconut variety was undertaken on the Illumina HiSeq 2000 platform, producing 40,367 transcripts. In addition, 14 known genes in somatic embryogenesis were discovered [127]. Sabana et al. (2018) examined coconut miRNAs using embryogenic calli transcriptome data, generating 27 unique mature miRNA sequences from 15 families. Conserved miRNAs play a role in phytohormone-mediated signaling and somatic embryogenesis pathways. This emphasizes the importance of subtle miRNA-mediated gene regulation in resolving in vitro recalcitrance in the coconut [128]. They profiled short RNAs in embryogenic and non-embryogenic calli and identified 110 conserved miRNAs, with 48 unique to embryogenic calli and 21 exclusive to non-embryogenic calli. This showed that conserved and species-specific miRNAs modulate SE in coconut and provide a resource for regulating the crop’s SE turnover [129]. Table 3 summarizes the data generated from the various coconut transcriptomics initiatives from the different tissues of different varieties of coconut mentioned above.

### 5.3. Transcriptomics for Fresh Insights on Defense Responses against Pathogen Attacks

Diseases caused by microbial pathogens impact coconut production. The major diseases include root (wilt) disease (RWD), caused by a soil-borne fungus known as “Phytophthora palmivora”, and coconut yellow decline (CYD) disease, caused by phytoplasma called “Candidatus Phytoplasma Palmae”, with significant reported losses in plantation in India and Malaysia, respectively. Muliyar et al., 2020, used comparative RNA-seq on the Illumina platform to study the global gene expression patterns of genes from the CGD’s healthy and diseased (RWD) palm. Differentially expressed genes (DEG) analysis found 2718 transcripts that were differentially expressed, with 136 transcripts unique to the diseased palm and 454 to healthy palms. As expected, proteins such as calmodulin-like 41 and WRKY DNA binding proteins, which are associated with plant–pathogen interactions, were overexpressed [95]. When host membrane receptors interact with effectors in the form of pathogen-associated molecular patterns (PAMPs), the basal defense response is triggered. Protein kinases are essential parts of these regulatory systems, affecting Ca2+ influxes and activating calcium signaling, while salicylic acid causes genes involved in defense responses, such as chalcone synthase, to be initiated in healthy palms with coordinated regulation by WRKY and NAC (NAM-ATAF1,2-CUC2) domain transcription factors [80].

Coconut yellow decline (CYD) is a devastating phytoplasma disease that affects the coconut. Nejat et al. (2009) reported that Malayan Red Dwarf (MRD) (Figure 2) coconut trees are more resistant to CYD disease than Malayan Tall and Yellow Dwarfs in a screening analysis of Malaysian coconut germplasm [130]. Nejat et al. (2015) used a comparative examination of healthy and infected MRD leaves to explore the activation of defensive mechanisms during the assault of yellow decline phytoplasma. There were 39,873 differentially expressed transcripts, with 18,013 upregulated and 21,860 downregulated in sick palm leaves compared to healthy ones. Most often, genes implicated in pathogen defense responses, particularly defense signaling, were elevated in diseased palms. Pathogenesis-related (PR) proteins, polyphenol oxidases, amino acid permeates, ABC transporters, cytochrome P450, flavonol synthases, and WRK transcripts that were downregulated were primarily involved in cytokinin production and photosynthesis. Altogether, the study revealed direct evidence for the activation of defense systems following phytoplasma assault in the MRD cultivar [131].

Verma et al. (2017) published the first transcriptome database of coconut root (wilt) disease (RWD) 2017. The database that was developed included candidate genes as well as SNPs and Indels. The database comprises 285,235 transcripts, biochemical pathways, transcription factors, and 22,021 DEGs. A total of 10126 and 97117 SSR markers were recovered from the DEGs and de novo transcriptome assemblies, respectively, and are a valuable genetic resource in producing and developing RWD-resistant cultivars [132].

Using RNA sequencing data from coconut root and leaf tissue exposed to SA, Silverio-Gómezand et al. (2022) discovered differentially expressed genes that play a role in the plant’s defense response. Consequently, 4615 unigenes with different expression levels were discovered in leaves and 3940 in roots. This analysis revealed functional categories relevant to the induction of defense response, such as “systemic acquired resistance,” as well as significantly enriched hormone categories, including abscisic acid. The findings revealed the most common antimicrobial compounds being produced based on KEGG pathway. The results support the hypothesis that exogenous SA injection activated pathogenesis-related genes (*PRs*), resistance gene analogs (*RGAs*), Isochorismate Synthas2 (*ICS2*), Non-Specific Lipid Transfer Protein 2 (*NLTP2*), Peroxidase 4 (*PER4*), Thioredoxin M (*TRXM*), and multiple WRKYs in plantlets through NPR1-dependent pathways [133]. Rajesh et al. (2015) found and described resistance genes in coconut plants resistant to coconut root (wilt) disease. The researchers used RNA-Seq to identify 243 resistance gene analog (RGA) sequences and divided them into six types. Based on TIR or CC motifs in the N-terminal regions, the phylogenetic analysis of deduced amino acid sequences indicated that coconut NBS-LRR type RGAs were categorized into various families. This work contributes to the knowledge of RGAs evolution in *Arecaceae* by providing a sequencing resource for creating RGA-tagged markers in coconuts to help locate disease-resistant candidate genes [134].

## 6. Conclusions and Future Perspective

The coconut is one of the most important tropical and subtropical crops, yet it has received less attention than other plants in the last few decades. However, there has been a leap in interest for coconut in recent years due to the unique physical, nutritional, and medicinal properties of coconut-derived ingredients suitable for food and cosmeceutical applications. Genetic improvement can potentially increase coconut production globally by focusing on certain traits contributing to yield. However, working with coconuts in greenhouses requires a large area of space to expand the benefits of the coconut breeding programs. A comprehensive and suitable protocol for coconut micropropagation would be very valuable. In addition, future research using advanced genome-assisted breeding technology that focuses on producing superior coconut cultivars that are high yielding and can withstand environmental perturbations are highly recommended. To this end, effective and economical ways to improve critical traits affecting yield and adaptability to diverse climatic conditions should be explored. 

The large-scale discovery of QTLs associated with yield and endosperm quality will have favorable consequences on improving copra and coconut oil production. Breeding perennial fruit trees, such as coconut, to increase fruit characteristics is difficult because of the long juvenile period and generation cycle, heterozygosity, detrimental biotic and abiotic factors, and lack of adequate genetic resources. MAS is more efficient and precise in choosing parental lines with the right genetic mix. Progeny with desirable or undesirable qualities may be identified at an early stage of development using the information gleaned from DNA testing and the parent’s past performance. Thus, it is important to tap into breakthroughs in biotechnological research that have been achieved in the last decade, such as the discovery of polymorphic molecular markers associated with particular desirable traits. The technique should be made simpler in practice for adoption by coconut breeders and farmers to improve productivity.

Whole genome sequencing and transcriptomic strategies on tall and dwarf coconut varieties have enabled the discovery and identification of molecular markers implicated in sugar and oil production and fruit growth and ripening as well as biotic and abiotic stresses that affects fruit quality. The availability of accessible genome sequences of coconuts has assisted in identifying SNP variants/indels, QTLs, and genes that may be used to identify the genetic contributors to fruit characteristics. The term “genome-wide selection” (GWS) or “genomic selection” (GS) refers to a method of selecting hundreds of DNA-based markers all at once, allowing for a dense coverage of the whole genome. In GS, the selection is performed using gene-based markers distributed across the whole genome, allowing for the precise prediction of complex traits. Additionally, a parent is chosen using a projected value known as genomic estimated breeding value (GEBV), determined using dense DNA markers throughout the whole genome. The genomic resources developed for coconut can be utilized to benefit from this advanced molecular breeding approach to improve precision and reduce time needed to achieve the desired objectives. This will enable the long-term sustainability of the coconut industry through molecular breeding.

The coconut has significant genetic diversity due to natural cross-pollination and centuries of cultivation in diverse geographical locations, allowing it to adapt to varied environmental circumstances. Yet, with the introduction of a number of high-yielding and genetically uniform cultivars in recent years, the coconut’s genetic diversity has declined. Little progress has been made regarding the description of coconut diversity compared to more commercially significant palms, the African oil palm (*Elaeis guineensis*) and date palm (*Phoenix dactylifera*). However, recent studies have shown tremendous variability in coconut populations utilizing molecular markers such as SNP [55,78] and microsatellites [51] (Sudha et al., 2023) Despite this progress, more research is needed to fully understand the diversity and distribution of coconut varieties to support conservation and breeding efforts.

## Figures and Tables

**Figure 1 plants-12-01913-f001:**
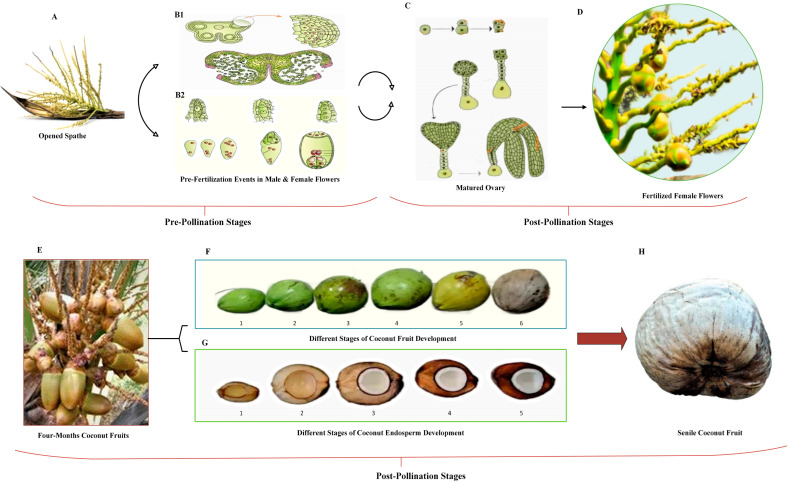
Pre-pollination and post-pollination stages in coconut fruit development. Pre-pollination stages: (**A**) opened coconut spathe [44]: the earliest differentiation of the ovaries from the female floral primordial occurs around six to seven months before the spathe opens. This is the stage at which the fruit begins to develop (initiation stages). (**B1**,**B2**) [45]: the pre-pollination stage, which spans from initiation to fertilization, starts to develop and expand in size by cell division. (**B1**): events before fertilization in the male flower (the formation of micro spores from pollen mother cell through meiosis, a process called microsporogenesis). (**B2**): events before fertilization in the female flower (megasporogenesis is a prosses that forms eight nucleated embryo sacs). Post-pollination stage: the growth factor in the post-pollination stage is the result of food accumulation, which causes cell expansion. This phase spans the period following fertilization until the coconut fruit reaches maturity. (**C**) [45]: the transfer of pollen from anther to stigma to produce a matured ovary is called pollination. (**D**,**E**) [46,47]: after fertilization, the mature ovary starts to grow and develop to produce the coconut fruit. (**F**,**G**) [48,49]: after the formation of the fruit, it begins to grow until it reaches maturity, which takes approximately 11–12 months for tall coconut varieties and 10–11 months for dwarf varieties. During this period, changes are observed in the size, shape, and color of the fruit and the content and shape of the endosperm. (**F**) (1) 4 months, (2) 5 months, (3) 6.5 months, (4) 8.5 months, (5) 10 months, and (6) 12.5 months. (**G**) (1) 3 months, (2) 6 months, (3) 9 months, (4) 11 months, and (5) 13 months coconut fruit. When the fruit reaches maturity, the outer surface (the exocarp) begins to brown. Certain types of fruits separate from the fruit stem and fall at this point. However, in some dwarf forms in particular, the ripe fruits do not fall off the palm even after the fruits have become totally brown. (**H**) [50]: in old coconut fruit, the endosperm within the nuts will dry up or decay if the fruits are not collected.

**Table 1 plants-12-01913-t001:** The strongest positive and negative effects of elite alleles related to coconut fatty acid content.

Trait	Marker	Locus	Genomic Position (Mb)	Phenotypic Effect Mean	Frequency	Carrier Germplasm	Reference
Fatty acid content	SSR-C-7566-3	CnfatB3-359	Chr11:12.36	+2.35	0.53	Aromatica Green Dwarf	[84]
Fatty acid content	SSR-C-7566-2	CnFatB2-830	Chr11:18.33	−0.65	0.15	MAWA	[84]

**Table 2 plants-12-01913-t002:** Statistic comparison of the four coconut genomes assembly.

Parameter (Variety)	Sequencing Platform	Estimated Genome Size (Gb)	Assembly Size (Gb)	Sequence Count	Scaffold N_50_ (Kb)	Scaffold Presenting Percentage of Genome	Seq Coverage	Contig N_50_ (Kb)	Reference
Hainan Tall	Ilumina HiSeq2000	2.42	2.2	1,11,366	418.07	90.91%	×173	72.64	[9]
Catigan Green Dwarf	PacBio-SMRT+Illumina Miseq	2.15	2.10	7998	151.93	97.6%	×175.91	46.36	[92]
Chowghat Green Dwarf	Ilumina HiSeq2000 +PacBioRSII	1.94	1.93	26,885	128.74	81.56%	×70.6	48	[95]
*Cn. Tall*	nanopore sequencing+Hi-C +Illumina HiSeq2000	2.42	2.4	34,251,876	-	83.55%	-	2927	[96]
*Cn. dwarf*		2.44	2.39	56,639,188	-	84.00%	-	14,296	

**Table 3 plants-12-01913-t003:** Statistics comparison of the six coconut transcriptome assembly.

Variety	Organs Used in Sequencing	Whole Nucleotide	Pair-End Reads	Clean Bases	Number of Unigenes	N50 Value of Unigenes	Total Transcripts	Average Length of Unigenes	Percentage of Gene Predicted from Genome-Seq Match to Transcriptome	Reference
Hainan Tall	8-month Endosperm and Leaf	4.9 (Gb)	54,931,406	Not Mentioned	57,304	1219	Not Mentioned	752	20,541 (71%)	[121]
Green Dwarf	8-month Embryo 5-month Endosperm Spear Leaf	10 (Gb)	81,128,552 103,080,366 121,151,552	12,435,008,760 9,682,020,834 13,609,335,965	58,211 61,159 33,446	951 969 912	86,254 229,886 159,509	732 684 744	24,857 (42.7%) 29,731 (48.6%) 26,069 (47.9%)	[124]
West Coast Tall	12-month zygotic embryo along with endosperm	7.73 (Gb)	50,839,994	5.42 (Gb)	73,308	561	161,426	436	40,367 (54.84%)	[127]
PLNT & LAGT	6–7-month Endosperm	4 (Gb)	395,511,022	226,653,479	416,488	874	267,827(PLNT) and 285,037 (LAGT)	406(PLNT) and 736 (LAGT)	436 (19%)	[122]
Green Dwarf	7-month endosperm (Aromatic and Non-Aromatic)	11 (Gb)	54,884,736 (Aromatic) and 55,031,246 (non-aromatic)	5,487,475,543 (Aromatic) and 5,502,110,501 (non-aromatic)	24,572 (Aromatic) and 23,322 (non-aromatic)	1074 (Aromatic) and 1134 (non-aromatic)	118,221 (Aromatic) and 95,613 (non-aromatic)	653 (Aromatic) and 662 (non-aromatic)	N	[107]
Baybay Tall (BAYT)	7-month Endosperm and Endocarp	66.9–88.9 (Mb)	586,631,950	334,181,119	95,474 (EC) and 103,979 (ES)	270–2287	234,588 (EC) and 284,696 (ES)	80.3–285.9	35,650 (28.1%)	[123]

## Data Availability

The data are contained within the article.

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
