# Peer review of "Genomics and Transcriptomics Reveal Genetic Contribution to Population Diversity and Specific Traits in Coconut"

_plants, 2023, doi:10.3390/plants12091913_

Round 1

Reviewer 1 Report

This is interesting review about the Cocos nucifera progres research using genomics, transcriptomics, and molecular markers initiatives. The topic presents interest because provide lots of current information about the genomic resources developed for coconut and this will enable the long term sustainability of the coconut industry through molecular breeding. In my opinion, this review can makes a significant contribution towards a better understanding of existing concepts about Cocos nucifera.

However, I have listed some suggestions for correction that the authors must consider. Thus, I would propose that the authors go once again through the text and correct all technical issues and more.

L. 26, 29, etc. The correct scientific name is Cocos nucifera, be careful please.

L. 83, 108, 244, 538. Please provide the references for these sentences.

L. 62, 128, 425, 719, 722, 738, etc. Write the scientific name in italics. Please check the entire manuscript.

L. 203-206. This sentence could not convey any meaning. The statement can be broken down into two smaller meaningful sentences. Please provide the reference for this sentence.

L. 211, 380. Remove the dot in front of the reference.

I suggest that in all the tables the reference number should be provided in the column intended for this purpose, so that the reader can quickly identify the reference in question. However, in my opinion, the References section is a bit limited, for a review paper. I recommend you to add more current references related to the topic of the manuscript.

Author Response

Reviewer 1

1- L 26,29 etc., The scientific name of coconut was corrected.

2- L 83, 108, 241, 560, References were added to these sentences.

3- All scientific names were written in italics.

4- l 205-208, Meaningless sentences were corrected and sources were added.

5- L 2013, 407. The extra point was removed.

6- All the tables were added to the reference number.

7- More references were added to the article.

Reviewer 2 Report

This manuscript submitted for revision (plants-2307881) provides a comprehensive review of the progress of coconut research using genomics, transcriptomics and molecular markers initiatives. This review paper is useful for breeding programs  for MAS selection to improve coconut genome and crop in the future. I have noticed some technical errors in the text, as follows: in line 197 before Earlier should be given "dot" (.). In line 198-9 the frase "restriction fragment length polymorphism" is written twice. In line '205 Kandoliya et al., 2018' sholud be given in bracket. In line 254 after [53] should be "dot". In line 308 "comma" is not needed before [60]. In line 380 "dot' is not needed before [63]. In line 274, 390 a space is needed before [43] and [73]. In line 354 the name of family Arecaceae should be written in italics. I recommend to publish this review article in Plants after minor technical correction.  

Author Response

Reviewer 2

1- L 200, A dot was added before earlier

2- L 201, One of the “restriction fragment length polymorphism” was removed

3- L 205-208, The intended reference (Kandoliya et al. 2018) was corrected

4- L 251, The dot was added after the bracket

5- L 311, The extra comma before the bracket was removed

6- L 407, The dot before the bracket was removed

  1. L 270, 417 Spaces were added before the brackets

8- L 366, Arecaceae family is italicized

Reviewer 3 Report

General: Language editing/correction is recommended!

Figure 1. Global Coconut Production 2000-2022  (FAOSTAT The figure should be omitted since it does not provide important or necessary information. Instead: note in the text that production was increased from approx. 50 mio t in the year 2000 to ca. 60 mio t in 2022

L 62ff    Coconut nucifera is the only surviving species in the genus Cocos. (…) Hybrid cultivars (Tall ×Dwarf) coconut have been produced by inter-specific crossing. Correct: Interspecific crosses have to be named.

Figure 2. Please mention origin/author(s) of the pictures/fotographs/drawings

Figure 3. Please mention origin/author(s) of the pictures/fotographs

Table 2. The whole table should be displayed on one page

L 696ff …discovery and identification of molecular markers implicated 697 in sugar and oil production, fruit growth and ripening as well as biotic and abiotic stress 698 that affects fruit quality.

It is recommended to add a table showing useful markers for major traits, their genomic position and contribution to explain the percentage of phenotypic vaiation explained by the individual markers!

L 710ff  It may be recommended to add a short paragraph commenting on the progress in describing diversity in the coconut in comparision to other major palm plants such as the oil palm.

Author Response

Reviewer 3

1- L 41-44, The figure of the global coconut production chart was removed and the desired information was added in the text.

2- L 64-68, Names of some hybrids and their interspecific crossings were added.

3- Sources of Figure 1 (L 170-190) and Figure 2 (550-551) were added

4- L 637, Table two was placed on one page.

5- L 325, Due to the fact that the genome-wide association study in coconut is very limited and the desired information was found only in one article, and the desired table was added in the Linkage Mapping, QTL Identification and Association Studies section.

6- L 735-744, The desired paragraph about genetic diversity was added in...

Others changes

1- All references were matched with the text and vice versa

2- The similarity ratio was reduced

3- The changes were displayed in blue color in the text

4- Sentences with green lines are related to the reduction of the similarity ratio.

5- Some more content was added to the text to increase the reference, which is shown in blue (L 384-398) and (298-306).